# Fractalkine Signalling (CX_3_CL1/CX_3_CR1 Axis) as an Emerging Target in Coronary Artery Disease

**DOI:** 10.3390/jcm12144821

**Published:** 2023-07-21

**Authors:** Shu Xian Loh, Yasemin Ekinci, Luke Spray, Visvesh Jeyalan, Thomas Olin, Gavin Richardson, David Austin, Mohammad Alkhalil, Ioakim Spyridopoulos

**Affiliations:** 1Department of Cardiology, Freeman Hospital, Newcastle upon Tyne NHS Foundation Trust, Newcastle upon Tyne NE7 7DN, UK; shuxian.loh1@nhs.net (S.X.L.); v.jeyalan@nhs.net (V.J.); mohammad.alkhalil@nhs.net (M.A.); 2Translational Research Institute, Vascular Biology and Medicine Theme, Faculty of Medical Sciences, Newcastle University, Newcastle upon Tyne NE1 7RU, UK; y.ekinci2@newcastle.ac.uk (Y.E.); luke.spray@nhs.net (L.S.); 3Academic Cardiovascular Unit, The James Cook University Hospital, Middlesbrough TS4 3BW, UK; david.austin@nhs.net; 4Population Health Science Institute, Newcastle University, Newcastle upon Tyne NE1 7RU, UK; 5Kancera AB, Karolinska Institutet Science Park, 171 65 Solna, Sweden; thomas.olin@kancera.com; 6Biosciences Institute, Vascular Biology and Medicine Theme, Faculty of Medical Sciences, Newcastle University, Newcastle upon Tyne NE1 7RU, UK; gavin.richardson@ncl.ac.uk

**Keywords:** acute myocardial infarction, atherosclerosis, inflammation, fractalkine, CX_3_CR1, monocytes, T lymphocytes, FRACTAL trial

## Abstract

Acute myocardial infarction (MI) is the most common and dramatic complication of atherosclerosis, which, despite successful reperfusion therapy, can lead to incident heart failure (HF). HF occurs when the healing process is impaired due to adverse left ventricular remodelling, and can be the result of so-called ischaemia/reperfusion injury (IRI), visualised by the development of intramyocardial haemorrhage (IMH) or microvascular obstruction (MVO) in cardiac MRI. Thus far, translation of novel pharmacological strategies from preclinical studies to target either IRI or HF post MI have been largely unsuccessful. Anti-inflammatory therapies also carry the risk of affecting the immune system. Fractalkine (FKN, CX_3_CL1) is a unique chemokine, present as a transmembrane protein on the endothelium, or following cleavage as a soluble ligand, attracting leukocyte subsets expressing the corresponding receptor CX_3_CR1. We have shown previously that the fractalkine receptor CX_3_CR1 is associated with MVO in patients undergoing primary PCI. Moreover, inhibition of CX_3_CR1 with an allosteric small molecule antagonist (KAND567) in the rat MI model reduces acute infarct size, inflammation, and IMH. Here we review the cellular biology of fractalkine and its receptor, along with ongoing studies that introduce CX_3_CR1 as a future target in coronary artery disease, specifically in patients with myocardial infarction.

## 1. Introduction

Cardiovascular disease (CVD) remains to this day the most common cause of death worldwide [1]. In Europe alone, an estimated 39 million deaths occur per year due to CVD, which accounts for 45% of all-cause mortality per year. Ischaemic heart disease (IHD) and stroke make up the majority of CVD in which IHD is still the single leading cause of death in Europe, despite its improvement in outcomes across the last three decades [2]. The estimated total cost incurred by CVD in Europe is about EUR 111 billion towards its health care alone; this does not even include the costs of productivity losses and informal care which can easily be associated with this disease.

## 2. Inflammation Is an Important Residual Risk Post Myocardial Infarction

Coronary artery disease (CAD) is the leading cause of mortality worldwide [3]. Its most serious manifestation, myocardial infarction (MI), leads to over 150,000 emergency admissions in the UK annually [4]. For patients following revascularisation in MI, secondary prevention now consists of targeting their residual risk, which is thought to be largely attributed to either hypercholesterolemia (treated with statins and PCSK9 inhibitors) or platelet aggregation (treated using dual antiplatelet therapy). Recently, inflammation, as quantified by hsCRP (high-sensitivity C-reactive protein), has been added to this list. The evidence for the role of inflammation is compelling. Among patients receiving contemporary statins, inflammation assessed by hsCRP is a stronger predictor for risk of future cardiovascular events and death than cholesterol [5]. In 2017, the CANTOS trial showed that reducing inflammation in patients with CAD and elevated hsCRP (>2 mg/L) improved outcomes [6]. In a more detailed sub-analysis of the CANTOS trial, the authors found that the relative improvement in outcomes correlated directly to the magnitude of hsCRP reduction [7]. Interestingly, a reduction below 1.8 mg/L reclassified patients into a lower risk group. A recently published TACTIC trial, which was a double-blinded randomised controlled pilot phase IIa study performed by our team, assessed the use of telomerase activator (TA-65) in reducing immune cell ageing in patients following MI [8,9]. The study observed that the average hsCRP in the placebo group was 3.9 mg/L even 12 months after ST elevation MI (STEMI), which would be considered as high risk. While myocardial infarction per se has been shown to accelerate the progression of coronary atherosclerosis even in the non-culprit artery [10,11], reducing inflammation is expected to delay this process. Our pilot data also suggest that different leukocyte populations have very different kinetics in the acute phase, compared to the chronic phase, post MI. Adverse remodelling is also propagated by excessive inflammation; early studies with the IL-1 antagonist Anakinra following MI suggest a reduction in heart failure [12]. To date, there is no longitudinal study in patients recovering with MI to phenotype the immune response in great depth over time, and correlating this to systemic inflammation. Moreover, the major limitation of the CANTOS trial was the effect of IL-1beta blockade on immunity, such that patients in the treatment group had a higher risk for infections, as well as a higher risk of dying from sepsis. There is, therefore, clearly a need for anti-inflammatory targets post MI without compromising immunity, or alternatively shorter duration of treatment with IL-1 or IL-6 antagonists.

## 3. Fractalkine Biology

### 3.1. Structure and Function of Fractalkine and Its Receptor

In 1997, scientists found a chemokine subclass which contained only one member in its subclass called fractalkine (FKN) or CX_3_CL1 on vascular endothelium [13,14]. A single fractalkine molecule consists of 373 amino acids in total (Figure 1). It can be broken down into four distinguished segments and is synthesised to span across a membrane, making it a unique transmembrane chemokine molecule [15]. The initial extracellular components of FKN consist of two separate parts which include an N-terminal domain, made up of 76 amino acids, which is attached to a mucin-like stalk containing a further 241 amino acids. The extracellular segment of the fractalkine molecule is then mounted onto a transmembrane α-helix consisting of 19 amino acids within the membrane space, before completing its model with an intracellular cytoplasmic tail that has its final 37 amino acids [15]. The extracellular segment of the fractalkine molecule can be cleaved by tumour necrosis factor-α converting enzyme (TACE or ADAM17) [16] and disintegrin-like metalloproteinase 10 (ADAM10) [17]. This creates a soluble component of fractalkine (sFKN), which allows for the chemoattraction of immune cells such as monocytes, NK cells, and T cells [18]. It is understood that ADAM10 organises for cleavage during periods of homeostasis [17], whilst ADAM17 is used to accelerate the process during periods of inflammation, stimulated by agents such as lipopolysaccharides (LPAs) or interleukin-1β (IL-1β) [16]. As fractalkine can be technically divided into two separate parts, it therefore allows a degree of versatility. The membrane-bound segment facilitates cell-to-cell adhesion of leukocytes to the endothelial cells, whilst the soluble segment acts as a chemoattractant to entice leukocytes to bind to the endothelial cells [19]. As endothelial cells act as the primary barrier that a migrating leukocyte will encounter, they are ready to support the function of the FKN molecule as its gatekeepers to the extravasation of leukocytes towards the site of inflammation [20]. However, bearing this in mind, FKN is not usually expressed in the absence of coronary artery disease. It is only when coronary arteries are diseased that FKN expression and upregulation are seen [21,22].

The fractalkine receptor, CX_3_CR1, was first identified by Harrison et al. in rats and was named initially RBS11 [24]. A year later, Combadiere et al. identified a similar gene in humans mapped to the chromosome 3p21 [25]. Both had identified CX_3_CR1 as a seven-domain transmembrane receptor coupled to a GTP-binding protein [20]. It is a highly selective surface receptor which is typically expressed on monocytes, CD4 T cells, CD8 T cells, NK cells, and γδ cells [15,26]. These cells would, in turn, react by stimulating the production of perforin and granzyme B. The interaction between fractalkine and CX_3_CR1 therefore contributes to the capture and transmigration of the desired leukocytes into the targeted area, resulting in the release of cytotoxic proteins and eventual endothelial and further tissue damage [18]. Interestingly, it is also found to be expressed by cardiomyocytes and fibrous tissue [27].

### 3.2. Classical Leukocyte Transmigration versus Fractalkine-Mediated Pathway

In order to understand the advantageous affinity of FKN, we first have to follow the classical method of leukocyte transmigration. It typically begins with leukocytes being captured and rolled on the endothelial surface with the help of E-, L-, and P-selectins [28]. The attachment of these immune cells to the endothelial surface then triggers the activation of integrins, which is used to secure the adhesion of leukocytes to the endothelium. This is further helped by interactions with endothelium-expressed ligand intracellular adhesion molecule (ICAM)-1 [29]. The trapped and captured leukocytes then exit the vascular wall and transmigrate into the affected tissue area [30]. In the presence of FKN, the high-affinity binding of FKN with CX_3_CR1 negates the need for integrin or selectin molecules as it no longer requires it for the adhesion of the leukocytes to the vascular wall [20,26]. As FKN has a high-affinity binding capability towards CX_3_CR1 receptors, it allows for faster binding, firmer adhesion, and bypass of the classical migration system to transmigrate leukocytes through its vascular wall [31]. It allows the cell to skip the rolling phase of leukocyte migration [32], hence replacing the need for VCAM-1 to integrin bindings, and is also thought to speed up the process of integrin-mediated transmigration of leukocytes-inflamed tissues [15,33,34].

### 3.3. Fractalkine/CX_3_CR1 Signalling

The specific signalling pathways generated by the association of FKN with CX_3_CR1 are still poorly understood. However, its FKN/CX_3_CR1 axis is known to affect multiple inflammatory signalling pathways including JAK-STAT, Toll-like receptor, MAPK, AKT, NF-κB, and Wnt/β-catenin, amongst many others [35]. One of the most commonly understood signals generated by the FKN/CX_3_CR1 pathway is the prevention of apoptosis, mainly in monocytes. Landsman et al. tested this theory by culturing monocytes with serum deprivation, and by means of exposing monocytes to a cytotoxic oxygenated cholesterol derivative called oxysterol [36]. A culture medium without soluble recombinant FKN demonstrated a higher percentage of dead monocytes compared to that with soluble recombinant FKN [36]. Moreover, the interaction between FKN/CX_3_CR1 induced the expression of anti-apoptotic genes from monocytes, mainly Bcl-2 and Bcl-xL (B-cell lymphoma extra-large) [36], strongly suggesting a survival effect of FKN signalling.

Inflamed vascular tissue expresses the transmembrane chemokine CX_3_CL1 (Figure 2), then capturing CX_3_CR1 positive leukocytes from the blood. Following the establishment of cellular contact, induced shedding of adhesion molecules facilitates the detachment of bound leukocytes from the endothelium. This could either lead to the release of these cells back into the blood flow or allow further migration of the leukocytes on and through the endothelium. Within the tissue, the leukocytes follow a chemotactic gradient of soluble chemokines to the site of inflammation.

## 4. Role of Fractalkine in Cardiovascular Disease

### 4.1. Role of Fractalkine and Its Receptor in Atherosclerosis

Atherosclerosis is the pathological basis for coronary artery, peripheral arterial, and cerebrovascular disease. It is a chronic inflammatory process affecting medium- to large-sized vessels which typically contain lipid-rich plaque lesions. Atherosclerosis is thought likely to be an immunomodulatory event, and therefore the prevalence of FKN or its receptor within different types of immune cells is highly relevant to the development of atherosclerosis. Wong et al. investigated coronary arteries in young patients who had died through acute trauma. They compared the presence of FKN staining in normal coronaries, with those from atherosclerotic and diabetic coronaries [21]. Surprisingly, FKN was not found within normal coronary arteries, but CX_3_CR1 was found to be diffuse in the cytoplasmic staining amongst the smooth muscle cells (SMCs). Comparatively, atherosclerotic coronary arteries demonstrated staining across the board in the intimal, medial, and adventitial layers for both FKN and CX_3_CR1. When this was compared to diabetic vessels, similar patterns of staining to atherosclerotic coronaries were found, with even heavier staining for FKN in the deep intimal layer demonstrated. This study was backed up when plasma FKN levels were compared between diabetics and non-diabetics, which demonstrated an increased level of FKN in diabetics compared to their non-diabetic counterparts [37]. Therefore, it is clear that FKN and CX_3_CR1 must play important roles in the pathogenesis of atherosclerosis and can be potentially used to assess cardiovascular risk in patients [38].

Whilst we understand from prior studies that atherosclerosis develops due to the infiltration of monocytes and T cells into the vessel wall [39], the mechanism for its cellular attraction has recently been proven by demonstrating that CX_3_CR1 gene-deleted apoE-/- mice had a significant reduction in macrophage infiltration towards the vessel wall as compared to apoE-/- mice with an intact CX_3_CR1 gene, when both murine groups were fed with a high-fat diet [40]. Similar findings were echoed in a subsequent study performed by Combadière et al. [41]. As gene silencing would be challenging in humans, controlling monocyte and macrophage action by means of pharmacological inhibition was demonstrated to provide similar and encouraging effects when tested again on rodents, as monocytosis was inhibited by blocking CX_3_CR1 [42]. All three of the above studies revealed that inhibition of CX_3_CR1 subsequently leads to prevention of atherogenesis. Whilst the systemic effect of treatment showed promising outcomes, Ali et al. studied the local impact of CX_3_CR1 by coating a prototype drug-eluting stent with a CX_3_CR1 antagonist [43]. The authors found reduced vascular inflammation and proliferation of SMCs in porcine models. This was compared to other polymer-coated stents without CX_3_CR1 antagonists as well as bare metal stents. The study demonstrated a 60% reduction in in-stent restenosis, with further reduction in peri-stent inflammation and accumulation of monocytes and macrophages, and minimal impact of endothelialisation. Hence, these studies demonstrate the proof of concept for the use of CX_3_CR1 antagonists in inhibiting neointimal hyperplasia. Nonetheless, there are two single nucleotide polymorphisms of the CX_3_CR1 molecule that have been identified and that contribute towards the risk of developing coronary artery disease, namely CX_3_CR1-V249I and CX_3_CR1-T280M [32]. CX_3_CR1-I249 allele heterozygosity was found to reduce the number of FKN binding sites, therefore actually reducing the risk of acute coronary events, and thus presents an independent genetic risk factor on its own [44]. Another study found a close association between CX_3_CR1-V249I and T280M polymorphisms, coronary plaque vulnerability, and acute MI, confirming that the subjects with the I249 allele display a low level of T cell inflammation and a reduced risk of developing vulnerable plaques [45]. Therefore, CX_3_CR1 is likely to amplify the Th1 immune response, directly related to plaque vulnerability, thus standing out as a potential target in fatal AMI prevention. This allows for the risk stratification methods using CX_3_CR1 gene coding to quantify the risk of coronary artery disease independent of other factors. In addition to leukocyte attraction, FKN expressed by endothelial cells can also activate and degranulate platelets that contain CX_3_CR1-corresponding receptors [46]. These platelets then trigger P-selectin surface expression, which then attach onto monocytes, allowing for more monocytes to transmigrate [47]. This complex formation can be a key aspect, especially when MI occurs as a consequence of atherosclerotic plaque rupture, causing platelet activation and aggregation [32].

### 4.2. Significance of Fractalkine and Its Receptor in Myocardial Infarction

As coronary atherosclerosis progresses further, it can become symptomatic, manifesting clinically as stable angina, or with acute coronary syndromes such as the emergency presentation of ST elevation myocardial infarction (STEMI). Several studies have looked at the relevance of sFKN as well as CX_3_CR1-expressing cells in the context of plaque rupture. Using intracoronary imaging methodology of either intravascular ultrasound (IVUS) or optical coherence tomography (OCT) to understand plaque morphology, studies have compared the levels of FKN and their corresponding receptors to the instability of plaque disease [48,49,50]. Patients with unstable angina and proven unstable plaque disease were found to have significantly higher levels of FKN and mononuclear cells expressing CX_3_CR1, as compared to stable angina patients or healthy controls. Hence, it can be surmised that FKN is involved in the pathogenesis of plaque vulnerability and consequently plaque rupture [49]. Additionally, Ikejima also demonstrated an increase in levels of monocytes, T lymphocytes, and NK cells, all of which express CX_3_CR1, and were seen to be elevated in the unstable angina cohort compared to other patient groups [48]. The evidence of FKN in the presence of a ruptured plaque had already been proven by the studies above; therefore, further investigations of more severe infarcts were performed. A study was performed to identify the different circulating proteins present in the event of an acute coronary syndrome, with the majority being STEMI patients. Helseth et al. identified a significant inverse relationship between total ischaemic time with soluble FKN (sFKN) levels where the cut-off was more or less than 4 h [51]. However, this study was not consistent in terms of timing of venepuncture as it was performed between 6 and 24 h of the PCI procedure with a median time of 18 h. Hence, an important study was subsequently performed by Njerve et al., which managed to demonstrate that in comparison to patients with stable angina, those with an acute MI were found to have statistically significant higher levels of FKN up to 12 h post procedure [52]. Levels then returned to those of patients with stable angina by 24 h and the reverse was seen in gene expression of CX_3_CR1 in patients with acute MI. However, Njerve failed to demonstrate any relationship between the levels of FKN and CX_3_CR1 expression with myocardial injury by means of troponin, CK-MB, or even infarct size and left ventricular ejection fraction (LVEF) measured on MRI scans. Another study demonstrated a higher level of sFKN in patients with acute STEMI as compared to stable angina, with a rapid decline in levels of sFKN within 24 h with successful reperfusion during PPCI [53,54]. Patients with an acute STEMI maintained a persistently higher level of sFKN compared to the stable angina group if they had not undergone the PPCI procedure. Moreover, the higher level of sFKN was positively correlated with the NT-proBNP level at 1 month, implying worse cardiac function. Our own studies showed that the level of sFKN dipped to its lowest point at 15 min post reperfusion and its level rapidly rose to a peak at 90 min [55]. This was coincidentally correlated with the lowest level of T-cell lymphopenia, which was shown to have a poorer prognosis for patients due to the larger development of microvascular obstruction (MVO). Aside from a demonstration of increased risk of developing poorer cardiac function, levels of FKN had also been proven to be prognostic for the likelihood of developing major adverse cardiovascular events (MACEs) in STEMI patients. Yao et al. revealed a positive correlation between the level of FKN day 1 post PPCI with the level of troponin at day 7 [53]. FKN levels were also found to be inversely proportional to the LVEF which was taken at 1 month, and that FKN was an independent predictor for MACE, whereby a higher FKN level at day 1 lead to an increased risk of MACE following a year’s follow-up period.

### 4.3. History of Myocardial Reperfusion Injury

A timely reperfusion of an occluded coronary artery is crucial to restore myocardial blood flow, as it is a prerequisite for myocardial salvage. In the event of total or partial coronary occlusion, this can cause a myocardial infarction. There are different methods to achieve reperfusion of the blocked artery, such as thrombolysis, emergency coronary angioplasty, or even, in extreme situations, coronary artery bypass grafting (CABG). These procedures are all performed with the intent of minimising myocardial cell death, preventing heart failure, and improving survival benefits [56]. However, the paradoxical evil of reperfusion itself has been demonstrated to cause harm to the myocardium in the form of ischaemia/reperfusion (IR) injury. This makes reperfusion therapy a double-edged sword towards the threatened myocardium of those who have been affected. The theory of IR injury was thought to have been introduced in the 1930s [57]. It was subsequently described by Jennings and his team in the 1960s [58,59]. Their study demonstrated an accelerated histological change in the canine’s myocardium causing myocardial cell death after subjecting canine hearts to a period of ischaemia via coronary ligation. Since then, as scientific experiments have progressed, there have been two reversible and two irreversible factors that have been postulated to contribute to the development of reperfusion injury, as shown in the Table 1 [57,60,61,62].

IR injury is considered significant because it contributes up to 50% of the final myocardial infarct size in animal models [60]. This also affects the degree of severity of left ventricular ejection fraction (LVEF) in a patient, subsequently leading to heart failure (HF). However, the exact mechanism and extent of IR injury in patients remains unclear.

### 4.4. Role of Fractalkine Signalling in Myocardial Reperfusion Injury

The duration of myocardial ischaemia following coronary artery occlusion is a key determinant of infarct size, and rapid reperfusion by PPCI improves clinical outcome [63,64]. However, even in patients with timely and successful stent PCI and normalised antegrade flow, imaging demonstrates failed myocardial reperfusion in 50% of patients, visible as MVO by cardiac MRI, which is an independent predictor of death and heart failure [65,66]. Severe reperfusion injury leads to intramyocardial haemorrhage (IMH), where disturbed vascular integrity and erythrocyte leakage accumulate iron-degradation products, which trigger persisting inflammation and fibrosis, resulting in adverse ventricular remodelling and, finally, heart failure [67]. Hence, two preventable therapeutic targets warrant clinical research: (a) prevention of reperfusion injury that would abrogate IMH, and (b) reduction in post MI inflammation to prevent adverse remodelling, thereby averting heart failure and death. We have published several studies to demonstrate a clear link between CX_3_CR1 and MVO, which, together with unpublished experimental studies in a rat MI model (unpublished data [68]), led us to initiate and perform the first pilot study (FRACTAL, ISRCTN 18402242) in humans using a small molecule allosteric inhibitor of CX_3_CR1 (KAND567) in 70 patients with acute anterior MI (please see below under Section 6). This ongoing phase IIa study is predicted to yield its first results by the second half of 2023. Given the promiscuity of all other chemokine receptors for multiple cytokines, CX_3_CR1 presents a unique advantage as an ideal drug target. Unlike most chemokines, fractalkine (CX_3_CL1) interacts with only one G-protein-coupled receptor, CX_3_CR1, which is expressed on platelets, smooth muscle cells, cardiomyocytes, monocytes, macrophages, natural killer cells, and T lymphocytes [31,69]. We have shown that fractalkine signalling links microvascular obstruction in acute MI with temporal changes in CX_3_CR1-expressing leukocyte subsets, including T cells and non-classical monocytes [55]. These clinical associations have led to animal studies together with our collaborators at Kancera, which show in the rat MI model that CX_3_CR1 inhibition prior to reperfusion limits infarct size by 50%, and reduces neutrophil influx, oedema, and finally IMH (unpublished data [68]). As myocardial infarction has been implicated to be one of the main causes of heart failure, and one of the independent factors contributing to this is myocardial ischaemia/reperfusion (I/R) injury, the role played by FKN/CX_3_CR1 has also been assessed. In vitro models of murine neonatal cardiomyocytes were subjected to a period of 3 h anoxia with an anaerobic buffer, and then subsequently reoxygenated for 2 h before being exposed to different quantities of sFKN [19]. The study demonstrated that FKN encouraged myocardial I/R by means of influencing expression of atrial natriuretic peptide (ANP), intracellular adhesion molecule-1 (ICAM-1), and matrix metalloproteinase-9 (MMP-9) which are all involved in causing cardiac dysfunction [19]. Neutralising FKN using a FKN antibody (TP233) leads to an improved heart function and pressure loading in the setting of experimental MI. Recent preclinical studies have implicated senescent cells, induced by ischemia/reperfusion (I/R), as a significant contributor to the inflammatory response and a source of FKN release [70,71,72]. Notably, intriguing findings suggest that the elimination of senescent cells post I/R decreased myocardial FNK expression, and was associated with notable benefits, including improved cardiac function and reduced scar size [70]. Moreover, the level of FKN being expressed correlated with the severity of heart failure expressed by the murine hearts. This hypothesis is consistent with the findings found in past and present studies looking at FKN/CX_3_CR1 as a pathogenesis for heart failure, irrelevant of the aetiology of the heart failure itself [27,73,74].

### 4.5. T Lymphocytes Become Activated in Heart Failure (HF) and Influence Cardiac Inflammation and Fibrosis

Murine models show that CD4 T cells, including regulatory T cells (T_reg_), are necessary to form a stable, fibrotic scar and prevent LV rupture following myocardial injury [75,76]. After the scar has formed, however, these cells continue to infiltrate the myocardium and their pro-fibrotic properties contribute to the progression to HF. Depletion of either CD4 or T_reg_ cells in established ischaemic HF reduces interstitial fibrosis and halts LV remodelling [77,78], and this is also seen in non-ischaemic heart failure models [79,80,81]. Conversely, adoptive transfer of T cells from mice after MI is sufficient to induce myocardial fibrosis and LV dysfunction in recipient, naïve mice [77,82,83]. Despite this strong evidence for pathogenicity in mice, it is not known whether T cells have the same pro-inflammatory, pro-fibrotic, harmful function in human HF—a translational gap. Immune checkpoint inhibitors (ICIs) have, however, highlighted T cells’ capacity to cause myocardial damage in non-HF patients. By disinhibiting T cells, often through programmed cell death protein 1 (PD-1) blockade, ICIs promote anti-cancer immune activity, but can also provoke fulminant myocarditis—as seen in PD-1 deficient mice [84,85]. Myocardial specimens from patients with ICI-induced myocarditis have abundant T lymphocytes and macrophages [86], and their peripheral blood contains clonally expanded populations of CD8^+^CCR7^-^CD45RA^+^ effector memory T cells (CX_3_CR1-expressing CD8^+^ T_EMRA_ cells) [87]. The mechanisms of T cell migration to the myocardium in chronic HF are also not fully understood. Our group has shown that fractalkine and its receptor, CX_3_CR1, are involved in recruiting effector memory T cells to the myocardium during MI [55], and our unpublished pilot data suggest this could also be true in HF; patients with reduced LVEF one year after MI display lower CX_3_CR1 fluorescence in effector memory T cells. We hypothesise this is due to increased CX_3_CR1/fractalkine interaction, as in vitro incubation of PBMCs with fractalkine reduces CX_3_CR1 fluorescence and, in vivo, MI patients with increased fractalkine concentration after reperfusion display reduced CX_3_CR1 receptor expression.

### 4.6. The Fractalkine Receptor CX_3_CR1 and Previous/Latent Cytomegalovirus Infection Accelerate Immune Ageing and Increases Cytotoxic T Cells in MI

Interestingly, as we and others have shown, latent cytomegalovirus (CMV) infection is also associated with clonal expansion of CD8^+^ memory T cells [88], myocardial T cell infiltration, and ventricular remodelling after MI [89]; whether CMV seropositivity contributes to subclinical myocardial inflammation in HF patients is unknown. We have previously shown that CMV-seropositive patients demonstrate signs of accelerated immune ageing following myocardial infarction, and this seems to link with impaired myocardial healing [90,91]. Importantly, in patients with previous CMV infection, where there is known abundance of virus-specific cytotoxic T lymphocytes, we found (i) an increased Th1 pro-inflammatory response, (ii) enhanced infiltration of the heart with T lymphocytes, and, finally, (iii) adverse cardiac remodelling [89,90,91,92,93]. Our own published work strongly suggests that an accumulation of a CX_3_CR1^+^, senescent T cell compartment with short telomeres may contribute to higher mortality and age-related myocardial decline, and predispose towards cardiovascular diseases [90,91,92,93]. Together, this indicates that cytotoxic CD4 and CD8 T lymphocytes could be instrumental in ongoing vascular as well as myocardial inflammation in reperfused MI patients. Therefore, identification of potential targets, such as CX_3_CR1, could provide ideal ground for anti-inflammatory immunotherapy, potentially avoiding progression of patients to future heart failure.

## 5. Targeting CX_3_CR1 with KAND567 (Kancera AB, Stockholm)

### 5.1. Mechanism

KAND567, previously coded as AZD 8797 HCl (Astra Zeneca) [94,95] (Figure 3), is a small synthetic molecule that is non-peptide based. It is a selective, non-competitive, allosteric antagonist of the receptor CX_3_CR1, which had been developed primarily to target inflammation in/during myocardial infarction and other cardiovascular conditions with an inflammatory component (Figure 4). KAND567 is currently the only fractalkine receptor antagonist which has been utilised specifically in cardiovascular disease. Nonetheless, it has been concurrently used to treat other non-cardiac inflammatory conditions such as multiple sclerosis, neuropathic pain, and acute pancreatitis. KAND567 binds to CHO-hCX_3_CR1 membranes with a K_d_ of 12 nM, whereas inhibition of adhesion under physiological flow conditions of a human B-lymphocyte cell line endogenously expressing human CX_3_CR1 is exerted with an IC50 of 5.7 nM.

### 5.2. Dose Rationale

The effects of KAND567 have been investigated in three separate studies in a rat model, where ischemia/reperfusion injury was established by the ligation of the left anterior descending (LAD) coronary artery for 30 min followed by reperfusion for 2 h. Findings demonstrated that treatment with KAND567 infused before the start of reperfusion resulted in a significant reduction in the infarction size vs. the control group. No reduction in infarct size was achieved when KAND567 was infused later during the reperfusion period (30 min after the start of reperfusion). Thus, treatment with KAND567 rescues cardiomyocytes at risk of ischemic death when administered acutely before the start of the reperfusion of the ischemic myocardium. KAND567 has furthermore been shown to markedly reduce peri-stent inflammation and monocyte/macrophage accumulation in a porcine model following metal stenting coated with polymer including KAND567 at a concentration of 1 μM [43]. Thus, resulting in a 60% reduction in in-stent coronary artery stenosis at 4 weeks following the intervention compared with bare metal stents as well as polymer-only coated stents. In different studies in atherosclerosis-prone LDL-receptor deficient mice on a high-cholesterol diet, KAND567 significantly reduced vascular macrophage infiltration by 50% and reduced intima media thickness. Furthermore, reduced plaque volume and a more stable plaque phenotype was noted following treatment with KAND567 [68]. KAND567 effectively inhibits key mechanisms of fractalkine-dependent leukocyte adhesion of human whole blood cells to endothelial cells at an IC_50_ of circa 300 nM. The relative potency (IC_50_) of KAND567 on CX_3_CR1-dependent cellular adhesion of human, porcine, or rat cells to endothelial cells supports the in vivo disease model data of acute and chronic inflammation, and translates into an effective concentration in man of 1:1 (porcine:human) or 4:1 (rat:man) to that observed in disease models. Pharmacological action of KAND567 was documented and studied from day 2 onwards. The ultimate target was to achieve a concentration of 1–2 μM, which was approximated at 2–4 x IC_50_ of KAND567 in primary human monocytes, T cells, and NK cells. Additionally, a phase IIa study of KAND567 in COVID-19 patients demonstrated proof of pharmacological action by an increase in plasma CX_3_CL1, when a dose of 250 mg twice a day (BID) was administered over a period of 7 days. This provided support for an effective concentration range in human whole blood of 0.3–1.2 μM and an effective dose of 200 mg three times a day (TID) or 250 mg BID as further supported by pharmacokinetics of KAND567. At 250 mg KAND567 BID, or 300 mg TID in healthy volunteers, the average concentration of KAND567 reached 1.9 μM and a significant decrease in surface density of CX_3_CR1 on NK, T cells, and monocytes was seen. However, it should be noted that the highest tolerated single ascending dose (SAD) of KAND567 was 2500 mg as identified in the phase I study, whilst 500 mg BID delivered across 7 days was the maximum tolerated multiple ascending dose (MAD). These were tolerability doses and did not pertain to efficacy. Both intravenous (IV) and oral (PO) preparations were chosen for administration in the FRACTAL study (see in detail under Section 6). The rationale for the initial administration of IV KAND567 was to allow for immediate perfusion of cardiac tissues prior to coronary reperfusion, targeting IR injury. In circumstances such as STEMI, PO administration may delay perfusion to the cardiac tissues due to exacerbating factors such as reduced gastrointestinal motility secondary to opioid use for pain relief (e.g., gastroparesis). Subsequent capsules administered following the completion of the IV infusion were intended to increase compliance post PCI, in addition to exploring the possible extension towards its use in an outpatient setting.

### 5.3. Safety and Tolerability

Data from experience of human exposure stem from three studies in healthy subjects. A total of 92 (62 subjects orally and 30 subjects intravenously) have been exposed to KAND567. KAND567 was well-tolerated and without treatment-related serious adverse events or clinically significant deviations in vital signs, ECG measurements, or laboratory parameters in healthy young and elderly female and male subjects following oral single ascending doses up to 2500 mg and following seven days of dosing up to 500 mg BID. These phase I studies also demonstrated no variation in bioavailability of KAND567 between the young and elderly cohort, with the elderly population being defined as being between the ages of 60 and 80 years. Hence, dosage remained the same between both groups.

The maximum tolerated dose was deemed to have been exceeded at 800 mg BID when four of six subjects after four to six days of dosing developed moderate to severe gastrointestinal adverse events and clinically significant increases in liver function tests (LFTs). These were fully reversible after cessation of dosing. Additional analysis of the liver findings could not explain the increase in LFTs (ALT) by effects on bile transporters, mitochondrial dysfunction, oxidative stress, or hepatocyte apoptosis/necrosis. Following both intravenous and oral administration of KAND567 to dog, there were no observations of changes in organ weight or histopathology in the liver in these animals, and liver enzyme changes were fully reversible following recovery. The clinically observed changes in bilirubin are therefore suggested to be related to the observed inhibition of bilirubin transporter and not hepatoxicity. No elevation was seen in inflammatory cytokines and MPO in plasma of healthy volunteers. Intravenous administration of KAND567 was well-tolerated and without treatment-related serious adverse events or clinically significant deviations in vital signs, ECG measurements, or laboratory parameters in healthy volunteers after 6 h intravenous infusions, yielding C_ss_ up to 1.8 μM. Prolonged intravenous application did lead to mild to moderate local thrombophlebitis in some cases, which was fully reversible after cessation of dosing.

### 5.4. Pharmakokinetics

KAND567 is rapidly absorbed with a C_max_ within 1–2 h after both single and multiple dosing. The terminal half-life ranges from 5.2 to 7.5 h. The oral clearance ranges from intermediate to low after a single dose and low after multiple dosing, suggesting KAND567 to be a low-clearance drug in man at steady state. Approximate dose linearity with respect to AUC was seen in the single dose and for the multiple doses at steady state.

### 5.5. Alternative Drugs to Target Fractalkine Signalling

To date, only a few anti-FKN drugs have been identified in the research market, including E6011 and JMS-17-2. The use of E6011 has been focused on assessing its effects on inflammatory conditions such as rheumatoid arthritis (RA) and Crohn’s disease [89,90,91,92,93,94], whilst JMS-17-2 was particularly emphasised for its effects as an anti-cancer medication [100,101]. Tanaka et al. had used a humanised IgG2 monoclonal antibody called E6011 as its anti-FKN pharmacological therapy [102]. They performed a phase II, double-blinded, placebo-controlled study using this drug on patients with RA and assessed the safety and efficacy of the drug. These patients had moderate to severe RA and had failed or had inadequate response to methotrexate (MTX) therapy. They were therefore randomised into a placebo group or an incremental dose of subcutaneous (SC) E6011 injection. Patients were mainly followed up for a 24-week period, with those who completed the 24 weeks entered into an extension phase, in which the patients were followed for a further 104 weeks. The extension phase patients received a further fortnightly injection of E6011 of 200 mg up to 102 weeks if they entered this phase [103]. The primary endpoint of the study was to evaluate if patients receiving the monoclonal antibody achieved American College of Rheumatology 20% improvement (ACR20) in response to the drug at 12 weeks. A total of 194 patients were randomised, and 169 patients completed the planned treatment. Although there were higher rates of improvement in the higher dosage arms at 12 weeks, it did not meet criteria for statistical significance. However, the study did demonstrate a statistically significant improvement at 24 weeks with a decrease in CD16^+^ monocyte count by week 2, irrespective of dosage of E6011 administered. It is therefore thought that E6011 functioned by inhibiting monocyte survival. Both short- and long-term assessment of safety also demonstrated the use of this monoclonal antibody to be generally safe [103,104,105].

Another team in Japan also looked at the safety and efficacy of E6011 in mild to moderate Crohn’s colitis patients by performing a multi-centre, open-label, phase 1/2 study [106,107]. These patients failed to respond to conventional therapy including anti-TNFα, and were therefore enrolled into the trial. Of the 28 patients who were enrolled, the study demonstrated E6011 to be safe and well-tolerated by Crohn’s patients. It also provided clinical improvement in symptoms in 56% of the patients who received treatment, with 16% achieving clinical remission. However, five (18%) patients were found to have developed anti-E6011 antibodies throughout the period of follow-up of the study.

JMS-17-2, on the other hand, is another form of fractalkine receptor antagonist which has mainly been studied in the pre-clinical stage. It utilises the pre-formed concept that breast cancer cells typically migrate out of the blood circulation with the aid of the chemokine receptor CX_3_CR1. Coupled with evidence of reduction in disseminated tumour cells at the skeletal level in fractalkine knockout mice [108], Shen et al. managed to synthesise a highly selective molecule called JMS-17-2 [100]. JMS-17-2 is a small molecule which inhibits the phosphorylation of ERK and functions in a dose-dependent fashion. The use of this molecule was proven to significantly reduce the migration of breast cancer cells in in vitro mice models by Shen and co-workers [100]. Additionally, pharmacokinetic assessment demonstrated a 60% reduction in disseminated tumour cells (DTCs) compared to the controlled models, and the majority of the animals were also found to be cancer-free.

Another study also invested in the use of JMS-17-2 as a modality for treating pancreatic ductal adenocarcinoma (PDAC) [101]. This was again an in vivo study which focused on assessing motility, invasion, and contact-independent growth of PDAC cells, which were typically upregulated in the presence of CX_3_CR1. The use of JMS-17-2 demonstrated promising results in impeding these effects by means of inhibiting AKT phosphorylation. Although this study is still in its infancy, it suggests pharmacological potential in targeting fractalkine in the devastating area of pancreatic cancer.

## 6. The FRACTAL Trial

The FRACTAL (FRACTalkine inhibition in Acute myocardiaL infarction) study was designed as a phase IIa, randomised, two-arm parallel-group, placebo-controlled, double-blind, multi-centre trial. It evaluated safety, tolerability, and anti-inflammatory and cardio-protective effects after intravenous and oral administration of KAND567 in ST-elevation myocardial infarction (STEMI) patients undergoing percutaneous coronary intervention. For this, a cohort of 70 STEMI patients were admitted into two large tertiary centres in the United Kingdom (Figure 5). Included were patients within 5 h of chest pain onset with new anterior ST segment elevation, and an angiographic image confirming an occlusion in the proximal or mid-left anterior descending coronary artery (LAD) with Thrombolysis in Myocardial Infarction (TIMI) flow of 0–2. Only patients between the ages of 18 to 75 years old were accepted into the study. Exclusion criteria included cardiogenic shock, previous MI in the LAD territory, documented previous left ventricular systolic dysfunction (ejection fraction < 40%), previous coronary artery bypass graft (CABG), current use of steroids, immunosuppression, or benzodiazepines, active malignancy, renal failure, hepatic failure, and patients with any contraindication to cardiac MRI (CMR) scanning. Arterial blood sampling was obtained at the start of primary percutaneous coronary intervention (PPCI). Following angiography and final confirmation of eligibility, patients were randomised to receive the intravenous (IV) bolus of either investigational medicinal product (IMP) or placebo. To avoid delivery of the IMP *after* reperfusion of the occluded vessel, percutaneous coronary intervention (PCI) was only commenced following the intravenous bolus of IMP/placebo. Patients then completed the remainder of their IMP infusion over 6 h. Following the completion of their intravenous infusion, patients in both arms continued an oral IMP/placebo regimen over 72 h. The primary outcome of this ongoing pilot study will be safety, with a number of key secondary outcomes being assessed in parallel. Blood samples at nine different time points will be used to assess any changes in the immune compartment as well as in inflammatory markers. Further secondary endpoints of the study include the cardioprotective effects of KAND567, as measured by gadolinium-contrast CMR. Every patient will receive a baseline CMR scan on day 3 post MI, followed by a subsequent scan after 3 months. Image analysis will be performed by a Core lab which is blinded to the treatment assignment.

## 7. Conclusions

In this state-of-the-art review we attempt to describe the therapeutic potential in targeting fractalkine and its receptor within cardiovascular disease, and more specifically in coronary artery disease. Several aspects of FKN biology make this a very attractive target for pharmacotherapy. FKN signalling appears to be critical in the interaction between cytotoxic leukocyte subsets, such as NK cells, non-classical monocytes, and certain memory T lymphocytes, all happening at the interface between endothelium and inflamed tissue. One would expect, therefore, limited consequences of FKN inhibition in healthy tissues, or immune suppression, such as with IL-1 or IL-6 inhibition, and a more focused diminution of the inflammatory reaction to MI or CAD in general. What points to a crucial signalling pathway in vascular diseases such as atherosclerosis, COVID-19-related vascular complications, and CMV-induced endothelial toxicity is the breadth of studies linking CX_3_CR1 with these conditions. Ongoing clinical trials, such as FRACTAL, will soon start to shed light on the therapeutic potential of fractalkine inhibition.

## Figures and Tables

**Figure 1 jcm-12-04821-f001:**
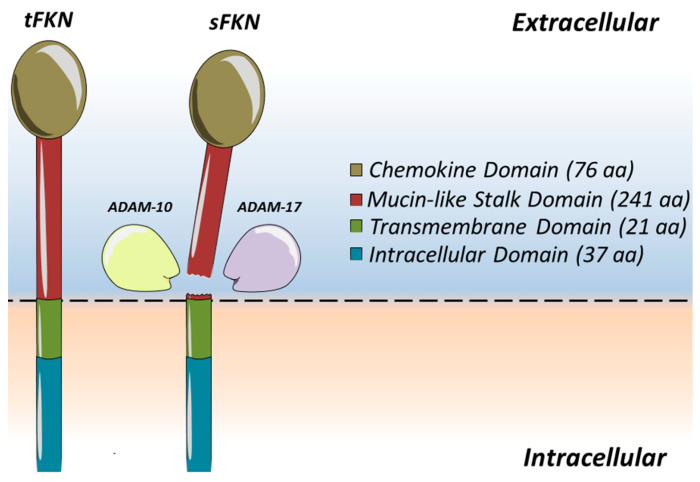
**Fractalkine (CX_3_CL1) is a transmembrane chemokine.** Fractalkine contributes to leukocyte binding and arrest on the vascular endothelium. Cleavage by the activity of the metalloproteinases ADAM10 or ADAM17 close to the cell membrane [16] of transmembrane fractalkine (tFKN) results in the release of the mucin-like stalk as well as the chemokine domain, generating a soluble secreted form (sFKN) that acts as a chemoattractant [23]. The remaining C-terminal cleavage fragment is thought to be removed from the cell membrane by intra-membranous cleavage by γ-secretase.

**Figure 2 jcm-12-04821-f002:**
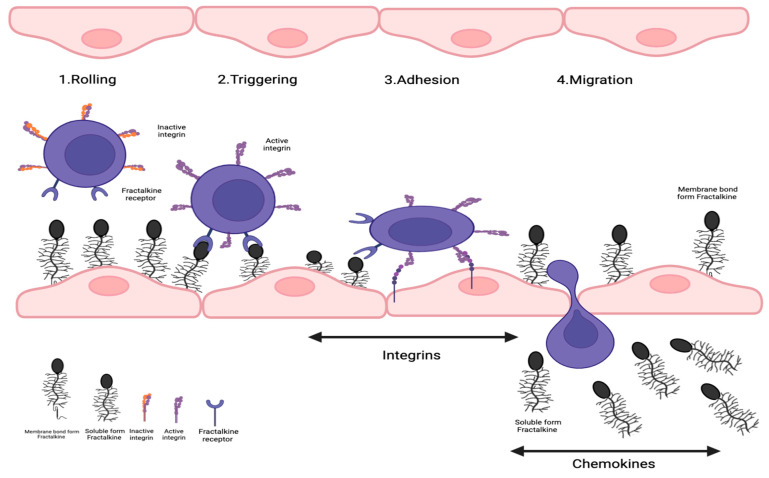
**Role of fractalkine in leukocyte recruitment in vascular inflammation.** Transmembrane chemokine CX_3_CL1 is expressed by inflamed vascular tissue, which then seizes CX_3_CR1-positive leukocytes from the circulation. Induced shedding of adhesion molecules would make it easier to separate bound leukocytes from the endothelium after the creation of cellular contact. This might cause the discharge of these cells back into the bloodstream or let leukocytes continue to move across and through the endothelium. Leukocytes would travel to the area of inflammation within the tissue by following a chemotactic gradient of soluble chemokines.

**Figure 3 jcm-12-04821-f003:**
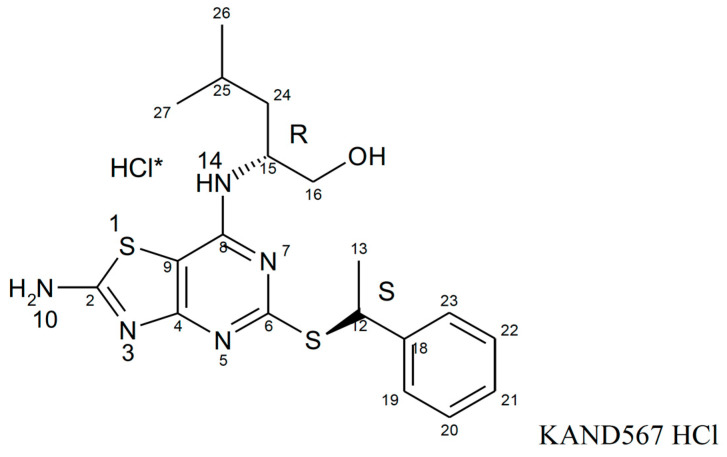
**Compound structure of KAND567.** * demonstrates that HCl is the salt component of KAND567 and is not a covalent part of the molecule. The HCl salt of KAND567 has a chemical name of (2*R*)-2-[(2-amino-5-{[(1*S*)-1-phenylethyl]sulfanyl}[1,3]thiazolo[4,5-*d*]pyrimidin-7-yl)amino]-4-methylpentan-1-ol hydro chloride.

**Figure 4 jcm-12-04821-f004:**
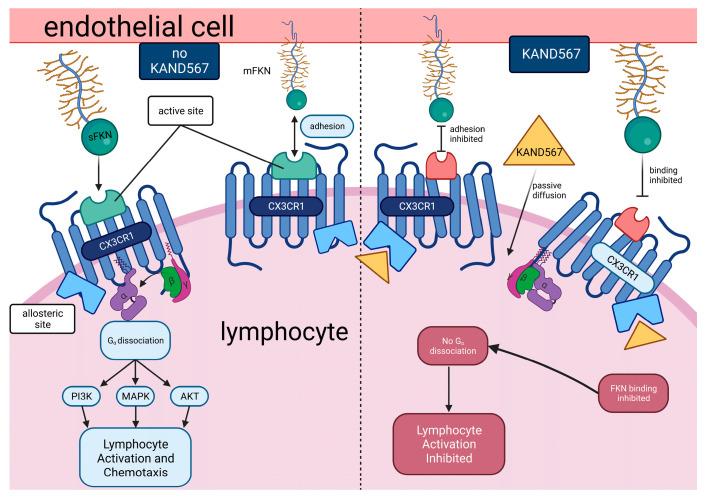
**Mechanism of KAND567-mediated inhibition of the CX_3_CR1 receptor.** KAND567 antagonises CX_3_CL1 binding and G-protein signalling by binding in an allosteric and non-competitive mode, intracellularly and close to the G-protein and beta-arrestin binding sites. As activating CX_3_CR1 triggers a cascade of multiple signalling pathways, one of the crucial pathways that KAND567 inhibits to provide its action is by inhibiting Ca^2+^ mobilisation [96,97] and Src activation to activate focal adhesion kinase (FAK) and to prevent cellular migration [98,99].

**Figure 5 jcm-12-04821-f005:**
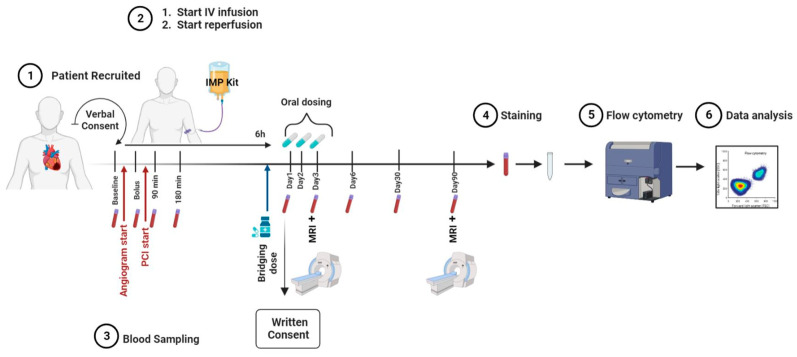
**Flowchart representing the setup of the time points for blood sampling and cardiac MRI**.

**Table 1 jcm-12-04821-t001:** **Reversible and Irreversible Factors of IR Injury**.

Reversible Factors	Irreversible Factors
Reperfusion arrhythmia	Microvascular obstruction (MVO)
Myocardial stunning	Lethal reperfusion injury

## Data Availability

Data sharing not applicable.

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
