# Peer review of "Fractalkine Signalling (CX3CL1/CX3CR1 Axis) as an Emerging Target in Coronary Artery Disease"

_jcm, 2023, doi:10.3390/jcm12144821_

Round 1
Reviewer 1 Report
In the present review, Shu et al. summarized the cellular biology of fractalkine and its receptor CX3CR1 within the cardiovascular disease, specifically in coronary artery diseases. They also introduce the allosteric antagonist of CX3CR1, KAND567, and associated clinical trials.
Overall, this review is comprehensive and logical. I only have a minor comment as follows.
1 The subtitle of 4.4 is not relevant to Fractalkine signaling.
2 Besides KAND567, there are other antagonists for CX3CR1, such as JMS-17-2. What is the rationale for only mentioning KAND567? Is it because KAND567 is the only drug to be evaluated in cardiovascular disease in a phase IIa trial? Please clarify this at the beginning of part 5 (Targeting Cx3CR1).
Author Response
- The subtitle of 4.4 is not relevant to Fractalkine signalling.
Thank you for your suggestion, we have decided to run with this section to demonstrate the relevance of T-cells and its importance in both ischaemic and non-ischaemic heart failure. Therefore, we do appreciate your comments, but this is the reason why we decided to continue with this section.
- Besides KAND567, there are other antagonists for CX3CR1, such as JMS-17-2. What is the rationale for only mentioning KAND567?Is it because KAND567 is the only drug to be evaluated in cardiovascular disease in a phase IIa trial? Please clarify this at the beginning of part 5 (Targeting Cx3CR1)
Thank you very much for your invaluable comments. We have made the appropriate alterations to the manuscript. (Changes are tracked on the attached revised manuscript)
“KAND567 is currently the only fractalkine receptor antagonist which has been utilized specifically in cardiovascular disease. Nonetheless, it has been concurrently used to treat other non-cardiac inflammatory conditions such as multiple sclerosis, neuropathic pain, and acute pancreatitis.”
“To date, there has only been a couple of anti-FKN drugs identified in the research market which includes E6011 and JMS-17-2. The use of E6011 has been focused on assessing its effects on inflammatory conditions such as rheumatoid arthritis (RA) and Crohn’s disease [89-94], whilst JMS-17-2 was particularly emphasized for its effects as an anti-cancer medication [100, 101].”
“JMS-17-2 on the other hand is another form of fractalkine receptor antagonist which has mainly been studied in the pre-clinical stage. It utilizes the pre-formed concept that breast cancer cells typically migrate out of the blood circulation with the aid of the chemokine receptor CX3CR1. Coupled with evidence of reduction in disseminated tumour cells at the skeletal level in fractalkine knockout mice [108], Shen et al. managed to synthesize a highly selective molecule called JMS-17-2 [100]. JMS-17-2 is a small molecule which inhibits the phosphorylation of ERK and functioned in a dose-dependent fashion. The use of this molecule had been proven to significantly reduce the migration of breast cancer cells in in vitro mice models by Shen and co-workers [100]. Additionally, pharmacokinetic assessment demonstrated a 60% reduction in disseminated tumour cells (DTC) compared to the controlled models, and majority of the animals were also found to be cancer free.
Another study had also invested in the use of JMS-17-2 as a modality for treating pancreatic ductal adenocarcinoma (PDAC) [101]. This was again an in vivo study which focused on assessing motility, invasion and contact-independent growth of PDAC cells, which were typically upregulated in the presence of CX3CR1. The use of JMS-17-2 demonstrated promising results in impeding these effects by means of inhibiting AKT phosphorylation. Nevertheless, this study is still in its infancy but suggests pharmacological potential in targeting fractalkine in the devastating area of pancreatic cancer.”
Reviewer 2 Report
Loh SX, J Clin Med, ms: 247168
The authors reviewed the consequences of acute myocardial infarction (MI) and possible therapies of the reperfusion-induced injury. It is noted that anti-inflammatory therapies carry the risk to affect the function of the immune system. The study includes the effects of fractalkine (FKN, CX3CL1), which is a chemokine, and present as a transmembrane protein on the endothelium, in connection with the corresponding receptor of CX3CR1. It is also presented that inhibition of CX3CR1 with an allosteric small molecule antagonist (KAND567) in the rat MI model, reduces the infarct size, inflammation processes and IMH. The authors review the cellular biology of fractalkine and its receptor, CX3CR1, as a future target in coronary artery disease, specifically in patients suffering from myocardial infarction.
The manuscript is well written and designed. However, the checking of English (e.g., grammatical errors and typos) is not the duty of the reviewer. The manuscript is evaluated on the scientific merit by this reviewer. This review manuscript is not a brand new summary, however, it gives a good and valuable picture about several signalling mechanisms, regarding MI and reperfusion-induced injury.
Comments and questions:
What would be the best formulation (technology) of fractalkine (e.g., i.v. infusion, pills, capules), which is a peptide structure compound?
What would be the optimal range of dosages of FKN in healthy volunteers and/or diseased patients? Is it the same range as in the case of KAND567 (max 2500 mg following 500 mg for a week) and/or TID or BID (200 mg and 250 mg, or 1 microM to 2 microM), respectively? Furthermore, What is the difference regarding the dosages between young, elderly male and female subjects? A table would be useful to include the dosages of the aforementioned age categories and interventions in the revised version of the review manuscript.
What could be the most important mechanism (number one), as a crucial signalling pathway, and therapeutic potential and fractalkine inhibition?
Finally:
The term of “reperfusion-induced injury”, including e.g., infarct size, necrosis, cardiac function, no-flow phenomenon, arrhythmias, and its mechanisms have been intensively studied starting from the 1980s and 1990s both under experimental and clinical conditions. Hundreds of mechanisms and interventions have been suggested to prevent reperfusion-induced injury. Therefore, it could be a valuable action, if some of the published papers (please, see the list below, back to the 80s) are acknowledged and cited in a separated subchapter, as the “history subchapter”, or incorporated into “4.2. Significance of Fractalkine and its Receptor in Myocardial Infarction” and/or “4.3. Role of Fractalkine Signalling in Myocardial Reperfusion Injury” in the revised version of the review manuscript (Loh SX, J Clin Med, ms: 247168).
The incorporation of suggested publications in the revised version of the manuscript may substantially increase the interest of general readers, senior and junior clinicians and experimental researchers.
Eur Heart J. 1983 May;4 Suppl C:49-54. doi: 10.1093/eurheartj/4.suppl_c.49
J Clin Invest. 1983 Sep;72(3):802-18. doi: 10.1172/JCI111051
-Basic Res Cardiol. 1984 Sep-Oct;79(5):562-71. doi: 10.1007/BF01910485
Circ Res. 1985 Feb;56(2):262-78. doi: 10.1161/01.res.56.2.262
Lucchesi BR, Werns SW, Fantone JC.J Mol Cell Cardiol. 1989 Dec;21(12):1241-51. doi: 10.1016/0022-2828(89)90670-6
-Circulation. 1989 Feb;79(2):441-4. doi: 10.1161/01.cir.79.2.44
-Eur J Pharmacol. 1989 May 19;164(2):293-302. doi: 10.1016/0014-2999(89)90470-6
-J Cardiovasc Pharmacol. 1990 Mar;15(3):398-407. doi: 10.1097/00005344-199003000-00009
Cardiovasc Res. 1992 Feb;26(2):101-8. doi: 10.1093/cvr/26.2.101
-Clin Biochem. 1993 Oct;26(5):359-70. doi: 10.1016/0009-9120(93)90112-j
J Clin Invest. 1993 Sep;92(3):1504-16. doi: 10.1172/JCI116729
Immunopharmacology. 1995 Feb;29(1):53-63. doi: 10.1016/0162-3109(95)00044-t
-Circulation. 1995 Oct 1;92(7):1866-75. doi: 10.1161/01.cir.92.7.1866
-Am J Physiol. 1996 Aug;271(2 Pt 2):H571-8. doi: 10.1152/ajpheart.1996.271.2.H571
-J Mol Cell Cardiol. 1996 Oct;28(10):2111-21. doi: 10.1006/jmcc.1996.0203
-Br J Pharmacol. 1997 Mar;120(6):1039-48. doi: 10.1038/sj.bjp.0701009
-Circulation. 1999 Apr 6;99(13):1685-91. doi: 10.1161/01.cir.99.13.1685
-Trends Cardiovasc Med. 1999 Nov;9(8):245-9. doi: 10.1016/s1050-1738(00)00029-3
-Ann N Y Acad Sci. 1999 Jun 30;874:412-26. doi: 10.1111/j.1749-6632.1999.tb09255.x
-Perfusion. 2004 Jul;19(4):207-19. doi: 10.1191/0267659104pf745oa
-Am J Physiol Heart Circ Physiol. 2004 Mar;286(3):H823-9. doi: 10.1152/ajpheart.00986.2003
-Can J Physiol Pharmacol. 2012 Sep;90(9):1185-96. doi: 10.1139/y2012-085. Epub 2012 Aug 22
-Circ Res. 2017 Mar 3;120(5):862-875. doi: 10.1161/CIRCRESAHA.116.310266. Epub 2016 Dec 8
Physiol Rev. 2019 Oct 1;99(4):1765-1817. doi: 10.1152/physrev.00022.2018
J Leukoc Biol. 2020 Sep;108(3):787-799. doi: 10.1002/JLB.2MR0220-549R. Epub 2020 Mar 17
Medicine (Baltimore). 2020 Sep 18;99(38):e22241. doi: 10.1097/MD.0000000000022241
-Tosaki A.Front Pharmacol. 2020 May 12;11:616. doi: 10.3389/fphar.2020.00616. eCollection 2020
J Am Heart Assoc. 2020 Oct 20;9(19):e017544. doi: 10.1161/JAHA.120.017544.
-Papanicolaou KN, Ashok D, Liu T, Bauer TM, Sun J, Li Z, da Costa E, D'Orleans CC, Nathan S, Lefer DJ, Murphy E, Paolocci N, Foster DB, O'Rourke B. J Mol Cell Cardiol. 2020 Feb;139:176-189. doi: 10.1016/j.yjmcc.2020.01.010. Epub 2020 Jan 29
-Exp Ther Med. 2022 Jun;23(6):430. doi: 10.3892/etm.2022.11357. Epub 2022 May 6
-J Am Heart Assoc. 2022 May 17;11(10):e024172. doi: 10.1161/JAHA.121.024172. Epub 2022 May 16
-Eur J Pharm Sci. 2023 Jun 1;185:106449. doi: 10.1016/j.ejps.2023.106449. Epub 2023 Apr 17
-Circulation. 2023 May 9;147(19):1444-1460. doi: 10.1161/CIRCULATIONAHA.122.060257. Epub 2023 Mar 29
-Biochem Biophys Res Commun. 2023 Mar 5;647:1-8. doi: 10.1016/j.bbrc.2023.01.049. Epub 2023 Jan 17
Author Response
- What would be the best formulation (technology) of fractalkine (e.g., i.v. infusion, pills, capules), which is a peptide structure compound?
Thank you for your question, we have added a paragraph to our review article with regards to this question as below.
“The use of both, intravenous (IV) and oral (PO) preparations, were chosen for administration in the FRACTAL study (see in detail under section 6.). The rationale for the initial administration of IV KAND567 was to allow for immediate perfusion of cardiac tissues prior to coronary reperfusion, targeting IR injury. In circumstances such as STEMI, PO administration may delay perfusion to the cardiac tissues due to exacerbating factors such as reduced gastrointestinal motility secondary to opioid use for pain relief (e.g. gastroparesis). Subsequent capsules administered following the completion of the IV infusion were intended to increase compliance post-PCI, in addition to exploring the possible extension towards its use in an outpatient setting.”
- What would be the optimal range of dosages of FKN in healthy volunteers and/or diseased patients? Is it the same range as in the case of KAND567 (max 2500 mg following 500 mg for a week) and/or TID or BID (200 mg and 250 mg, or 1 microM to 2 microM), respectively? Furthermore, What is the difference regarding the dosages between young, elderly male and female subjects? A table would be useful to include the dosages of the aforementioned age categories and interventions in the revised version of the review manuscript.
Thank you for your question, we have extended our explanation to our dosage rationale in the section under Dose rationale as below.
“Pharmacological action of KAND567 was documented and studied from day 2 onwards. The ultimate target was to achieve a concentration of 1-2 mM, which approximated at 2-4 x IC50 of KAND567 in primary human monoctes, T-cells and NK cells. Additionally, a phase IIa study of KAND567 in Covid patients demonstrated proof of pharmacological action by increase in plasma CX3CL1, when a dose of 250 mg twice a day (BID) was administered over a period of 7 days. This provided support for an effective concentration-range in human whole blood of 0.3 – 1.2 mM and an effective dose of 200 mg three times a day (TID) or 250 mg BID as further supported by pharmacokinetics of KAND567. At 250 mg KAND567 BID, or 300 mg TID in healthy volunteers, the average concentration of KAND567 reached 1.9 mM and a significant decrease in surface density of CX3CR1 on NK, T-cells and monocytes was seen. However, it should be be noted that the highest single ascending dose (SAD) of KAND567 tolerated was 2500 mg as identified in the phase I study, whilst 500 mg BID delivered across 7 days was the maximum tolerated multiple ascending dose (MAD). These were tolerability doses and not pertaining to efficacy.”
- What could be the most important mechanism (number one), as a crucial signalling pathway, and therapeutic potential and fractalkine inhibition?
Thank you for your question, we have added out most important mechanism to note with KAND567 under Figure 4.
“As activating CX3CR1 triggers a cascade of multiple signalling pathways, one of the crucial pathways that KAND567 inhibits to provide its action is by inhibiting Ca2+ mobilization [96, 97] and Src activation to activate focal adhesion kinase (FAK) and to prevent cellular migration [98, 99]”
- The term of “reperfusion-induced injury”, including e.g., infarct size, necrosis, cardiac function, no-flow phenomenon, arrhythmias, and its mechanisms have been intensively studied starting from the 1980s and 1990s both under experimental and clinical conditions. Hundreds of mechanisms and interventions have been suggested to prevent reperfusion-induced injury. Therefore, it could be a valuable action, if some of the published papers (please, see the list below, back to the 80s) are acknowledged and cited in a separated subchapter, as the “history subchapter”, or incorporated into “4.2. Significance of Fractalkine and its Receptor in Myocardial Infarction” and/or “4.3. Role of Fractalkine Signalling in Myocardial Reperfusion Injury” in the revised version of the review manuscript (Loh SX, J Clin Med, ms: 247168).
Thank you for your invaluable advise. We have made the appropriate changes as below.
“4.3. History of Myocardial Reperfusion Injury
A timely reperfusion of an occluded coronary artery is crucial to restore myocardial blood flow, as it is a prerequisite for myocardial salvage. In the event of total or partial coronary occlusion, this can cause a myocardial infarction. There are different methods to achieve reperfusion of the blocked artery, such as thrombolysis, emergency coronary angioplasty or even in extreme situations coronary artery bypass grafting (CABG). These procedures are all performed with the intent of minimising myocardial cell death, preventing heart failure and improving survival benefits [56]. However, the paradoxical evil of reperfusion itself has been demonstrated to cause harm to the myocardium in the form of ischaemia / reperfusion (IR) injury. This makes reperfusion therapy a double-edged sword towards the threatened myocardium of those who have been affected. The theory of IR injury was thought to have been introduced in the 1930s [57]. It was subsequently described by Jennings and his team in the 1960s [58, 59]. Their study demonstrated an accelerated histological change in the canine’s myocardium causing myocardial cell death after subjecting canine hearts to a period of ischaemia via coronary ligation. Since then, as scientific experiments have progressed, there have been two reversible and two irreversible factors that have been postulated to contribute to the development of reperfusion injury, as shown in the table below [57, 60–62].
|
Reversible Factors |
Irreversible Factors |
|
Reperfusion arrhythmia |
Microvascular obstruction (MVO) |
|
Myocardial stunning |
Lethal reperfusion injury |
The reason why IR injury is considered significant is because it contributes up to 50% of the final myocardial infarct size in animal models [60]. This would also affect the degree of severity of left ventricular ejection fraction (LVEF) in a patient, subsequently leading to heart failure (HF). However, the exact mechanism and extent of IR injury in patients remains unclear.”